# Compositional Generalization in Unsupervised Compositional Representation Learning: A Study on Disentanglement and Emergent Language

**Zhenlin Xu**    **Marc Niethammer**    **Colin Raffel**
Department of Computer Science
University of North Carolina at Chapel Hill
{zhenlinx, mn, craffel}@cs.unc.edu

## Abstract

Deep learning models struggle with compositional generalization, i.e. the ability to recognize or generate novel combinations of observed elementary concepts. In hopes of enabling compositional generalization, various unsupervised learning algorithms have been proposed with inductive biases that aim to induce compositional structure in learned representations (e.g. disentangled representation and emergent language learning). In this work, we evaluate these unsupervised learning algorithms in terms of how well they enable *compositional generalization*. Specifically, our evaluation protocol focuses on whether or not it is easy to train a simple model on top of the learned representation that generalizes to new combinations of compositional factors. We systematically study three unsupervised representation learning algorithms – $\beta$-VAE, $\beta$-TCVAE, and emergent language (EL) autoencoders – on two datasets that allow directly testing compositional generalization. We find that directly using the bottleneck representation with simple models and few labels may lead to worse generalization than using representations from layers before or after the learned representation itself. In addition, we find that the previously proposed metrics for evaluating the levels of compositionality are not correlated with the actual compositional generalization in our framework. Surprisingly, we find that increasing pressure to produce a disentangled representation (e.g. increasing $\beta$ in the $\beta$-VAE) produces representations with *worse* generalization, while representations from EL models show strong compositional generalization. Motivated by this observation, we further investigate the advantages of using EL to induce compositional structure in unsupervised representation learning, finding that it shows consistently stronger generalization than disentanglement models, especially when using less unlabeled data for unsupervised learning and fewer labels for downstream tasks. Taken together, our results shed new light onto the compositional generalization behavior of different unsupervised learning algorithms with a new setting to rigorously test this behavior, and suggest the potential benefits of developing EL learning algorithms for more generalizable representations.

## 1   Introduction

A human's ability to recognize or generate novel combinations of seen elementary concepts, also known as *compositional generalization*, is desirable for building general artificial ingelligence (AI) systems [22, 16, 3]. The Recognition-By-Components theory by Biederman [3] influenced the early development of computer vision models that are inherently compositional, e.g., hierarchical features [13, 14] and part-based models [37, 38]. However, modern deep learning systems still struggle with this key capability of human intelligence [33]. A few works studied specific spatial and

36th Conference on Neural Information Processing Systems (NeurIPS 2022).

object-wise compositionality [42, 30] or more general compositionality in the space of pre-defined attributes [45, 39] or the semantics of human language descriptions [49, 44].

Humans often express complex meaning in a compositional manner: we combine elementary representations to describe observations. For example, an object with simple geometry is described by separate and independent properties such as color (red, blue, ...), position (left, right, close, far away,...), and shape (circle, triangle, ...). Therefore, *compositional representations* are thought as helpful or even essential to achieve compositional generalization [22, 15, 3, 28]. However, considering that there are exponentially many possible combinations of a given set of elementary concepts, we need to deal with this combinatorial explosion for real-world visual observations. It is unrealistic to annotate enough data to learn the fine-grained compositionality. Therefore, unsupervised compositional representation learning is appealing because it does not require comprehensive labeling. However, unsupervised representation learning heavily relies on the design of an effective inductive bias (e.g. on the representation formulation) to induce the emergence of compositional representations.

A widely explored representation formulation with explicit compositionality is *disentanglement*. The most common formulation of disentanglement is that the generative factors of observations should be encoded into different factors of low-dimensional representations, and a change of a single factor in an observation leads to a change in a single factor of the representation. State-of-the-art unsupervised disentanglement models [20, 24, 27, 7, 34] are largely built on top of variational generative models [25]. To measure the level of disentanglement, various quantitative metrics have been proposed that are defined based on the statistical relations between the learned representations and ground truth factors with an emphasis on the separation of factors. A summary of disentanglement metrics and methods is provided in [31]. Another representation learning approach with an inductive bias towards compositionality is *emergent language learning*. Natural language allows us to describe novel composite concepts by combining expressions of their elementary concepts according to grammar. Therefore, linguists have been interested in studying the compositionality of discrete codes evolving during multi-agent communication when agents learn to complete a task cooperatively [9, 29, 19, 40]. Compositionality metrics with language structure assumptions, e.g. topographic similarity [4], were used to evaluate the learned language. However, for the above two types of methods, relatively few studies [36, 41, 6, 1] have directly evaluated how well the learned representations generalize to novel combinations on downstream tasks, which is the main motivation for compositional representation learning in the first place.

In this work, we study the *compositional generalization* performance of representations learned from unsupervised learning algorithms with two types of inductive biases for compositionality: disentanglement and emergent languages (EL). Instead of measuring the compositionality and disentanglement metrics defined based on various assumptions, we directly measure the generalization performance on novel input combinations with a two-stage protocol. Specifically, with a dataset divided into train and test sets ensuring that the test set contains *novel combinations of concepts* that never appear in the train set, we first learn an unsupervised representation model from unlabeled images in the train set. With very few labeled samples from the train set and the frozen unsupervised representation model, we train simple (e.g. linear) models on top of learned representations to predict the ground truth value for each generative factor of the dataset and evaluate these simple models on the test set. These choices are aligned with common practices in recent deep representation learning works, for example, self-supervised representation learning [10] and semi-supervised learning with generative models [26]. Different from previous studies (e.g. [6, 36]) that measure unsupervised learning performance (e.g. image reconstruction), we evaluate the performance on downstream tasks. We also emphasize how easily we can obtain downstream task models with the learned representation, e.g. when using very few labels and simple linear models. These designs highlight the generalization performance of the unsupervised learning stage, different from a setup that uses many more or even all labeled samples of the train set in the downstream task learning stage [10, 41] or performs unsupervised learning on the entire dataset [31]. More importantly, we study not only the compositional generalization of intermediate representations at the model bottleneck, e.g. where the disentangled latent variables are formulated, but also the representations from layers before or after it.

With the above evaluation protocol for compositional generalization, we explore selected unsupervised learning algorithms by varying (1) the hyperparameters of each algorithm that may control the levels of compositionality; (2) image datasets and the amount of data for both unsupervised and supervised learning stages; and (3) design choices of EL learning. First, we find that, compared to the low-

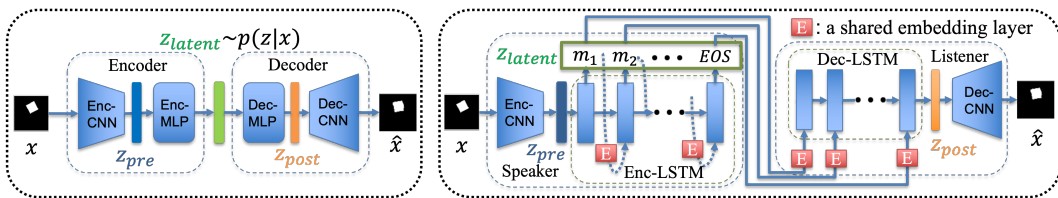

Figure 1: Architectures of disentanglement models (left) and emergent language models (right).

dimensional latent variables from the model bottleneck, representations from the layers before or after the model bottleneck enable better compositional generalization. Second, we find that attaining higher scores on previously proposed compositionality/disentanglement metrics does not always correlate with better generalization performance. Finally, the representations learned from EL models show stronger generalization performance than disentanglement models. These findings reveal the divergence between the efforts to achieve better compositionality / disentanglement metric scores and the initial motivation to obtain better generalization performance. To our best knowledge, this is the first study to compare representations in disentanglement models and emergent language learning through the lens of compositional generalization with a unified evaluation setup. We also discuss the advantage of the emergent language representation format versus disentanglement and connect it with recent related research in representation learning.

## 2 Unsupervised Learning with Compositional Representation Inductive Bias

In this section, we introduce more details on unsupervised learning algorithms with two different compositionality-seeking inductive biases on the representation formulation: disentangled representations in Section 2.1 and emergent languages in Section 2.2.

### 2.1 Learning Disentangled Representations

The concept of disentanglement assumes that high-dimensional observations of the real world $\mathbf{x}$ can be represented by low-dimensional latent variables $\mathbf{z}$, where each dimension of $\mathbf{z}$ encodes independent factors of variations in $\mathbf{x}$. For unsupervised disentanglement learning algorithms, we select the popular $\beta$-VAE and $\beta$-TCVAE methods which both modify the evidence lower bound (ELBO) objective in the variational autoencoder (VAE).

$\beta$-VAE use a hyperparameter $\beta$ for the Kullback-Leibler (KL) regularization term of the vanilla VAE loss to control the bandwidth of the VAE bottleneck:

$$\mathbb{E}_{p(\mathbf{x})}[\mathbb{E}_{q_\phi(\mathbf{z}|\mathbf{x})}[\log p_\theta(\mathbf{x}|\mathbf{z})] - \beta \mathbf{KL}(q_\phi(\mathbf{z}|\mathbf{x})||p(\mathbf{z}))], \tag{1}$$

where $p(\mathbf{z})$ is the assumed prior distribution of the latent variables and its conditional distribution $q_\phi(\mathbf{z}|\mathbf{x})$ is parameterized by a neural network (encoder) whose parameters are $\phi$ and the posterior $p_\theta(\mathbf{x}|\mathbf{z})$ is parameterized by a decoder whose parameters are $\theta$. $\beta = 1$ corresponds to the VAE loss.

$\beta$-TCVAE further decomposes the KL term in Eq. (1) into mutual information, total correlation, and dimension-wise KL terms, and penalizes the total correlation with the hyperparameter $\beta$:

$$\mathbb{E}_{q(\mathbf{z}|\mathbf{x})p(\mathbf{x})}[\log p_\theta(\mathbf{x}|\mathbf{z})] - \alpha I_q(\mathbf{x};\mathbf{z}) - \beta \mathbf{KL}(q_\phi(\mathbf{z})|| \prod_j q_\phi(z_j)) - \gamma \sum_j \mathbf{KL}(q(z_j)||p(z_j)), \tag{2}$$

where $I_q(\mathbf{x};\mathbf{z})$, $\mathbf{KL}(q_\phi(\mathbf{z})|| \prod_j q_\phi(z_j))$ and $\sum_j \mathbf{KL}(q(z_j)||p(z_j))$ are the mutual information term, total correlation term and the dimension-wise KL term respectively. The proposed $\beta$-TCVAE uses $\alpha = \gamma = 1$ and tunes $\beta$ only.

### 2.2 Learning Emergent Language

An alternative compositional representation learning method is emergent language (EL) learning, which aims to learn a representation that mimics the properties of natural language. The emergent language consists of sequences of discrete symbols from a vocabulary. Since the model combines discrete symbols in the vocabulary to represent complex semantics in observations, one expects that

meaningful compositionality might naturally emerge in communication between multiple agents using the emergent language to solve tasks. We consider the typical speaker-listener model (two-agent communication) for EL learning, as shown in Fig. 1. We apply EL learning to the image reconstruction task to be consistent with the reconstruction objective used by variational auto-encoder-based models. Note that in our setting the terminology "speaker-listener" is equivalent to the more common "encoder-decoder" terminology. The task is as follows.

1. The speaker receives an input $\mathbf{x}$ and encodes it as a message $\mathbf{m} = \{m^1, m^2, ...\}$, a sequence of discrete symbols from the vocabulary $V = \{c_1, c_2, c_{n_V}\}$ of size $n_V$. The maximum length of $\mathbf{m}$ is $n_{\mathrm{msg}}$.

2. The listener model receives the message $\mathbf{m}$ and the outputs $\hat{\mathbf{x}}$, aiming to accurately reconstruct the encoder input $\mathbf{x}$.

In this work, we use a speaker and a listener that are both hybrids of a convolutional neural network and an LSTM recurrent neural network. The flattened convolutional embedding of the input image, $\mathrm{EncConv}(\mathbf{x})$, is used as the initial cell state of an LSTM encoding module (EncLSTM). EncLSTM generates a discrete distribution $q(m^t|x)$ over $V$ at each time step $t$ autoregressively (with the embedding of the discrete token sampled in the previous step $t-1$ as input):

$$q(m^t|\mathbf{x}) = \mathrm{EncLSTM}(m^{t-1}|\mathrm{EncConv}(\mathbf{x}), \mathrm{emb}(m^1), .., \mathrm{emb}(m^{t-2})), \qquad (3)$$

where $\mathrm{emb}(\cdot)$ is the learnable layer that projects a discrete token into a high-dimensional embedding. We use the Gumbel-Softmax [23, 32] to sample from the discrete distribution $q(m^t|x)$ and the "straight-through" (ST) gradient estimator [2] for quantization (from soft-distribution to one-hot vector). This allow us to estimate the gradients from the discrete sampling process.

$$m^t = \mathrm{ST}(\mathrm{GumbelSoftmax}(q(m^t|\mathbf{x}))). \qquad (4)$$

To allow the message to be of variable length, which better mimics natural language, we set one token in $V$ to be the end-of-sequence (EOS) token that indicates the message end.

When the listener decodes the message $\mathbf{m}$, it first maps each discrete symbol into an embedding based on a learnable embedding layer and uses a decoding LSTM layer (DecLSTM) to process the sequence of embeddings recurrently.

$$\mathrm{Emb}^t(\mathbf{m}) = \mathrm{DecLSTM}(\mathrm{emb}(m^t)|\mathrm{emb}(m^1), \mathrm{emb}(m^2), .., \mathrm{emb}(m^{t-1})). \qquad (5)$$

We use the output of DecLSTM at the ending step $T$ as the embedding of the message to be the input of a convolutional decoder for image reconstruction:

$$\mathrm{Emb}(\mathbf{m}) = \mathrm{Emb}^T(\mathbf{m}), \text{where } T = \min(n_{msg}, \underset{i}{\arg\min}\{m^i == EOS, i \in \{1..N\}\}). \qquad (6)$$

Finally, the convolutional decoder (DecConv) reconstructs the input by:

$$\hat{\mathbf{x}} = \mathrm{DecConv}(\mathrm{Emb}(\mathbf{m})). \qquad (7)$$

## 3 Experimental Design

### 3.1 Datasets

We are interested in whether representations can generalize to novel combinations of seen concepts. Therefore, we need datasets that provide ground-truth labels of elementary concepts for (1) creating train/test splits and (2) downstream task evaluation. We consider datasets with $n_{gen}$ independent generative factors $F = \{f_1, ..., f_{n_{\mathrm{gen}}}\}$ where the space of factor $f_i$ is $S_i$. For example, if $f_1$ is color, then $S_1$ could be $\{yellow, red, blue, ...\}$. The data space $D$ is defined by the Cartesian product of the spaces of each factor, and therefore the cardinality of the dataset is $|D| = \prod_i^{n_{\mathrm{gen}}} |S_i|$.

In our study, we used two public image datasets studied in the disentanglement literature: dSprites [35] and MPI3D [17]. The dSprites dataset contains images of 2D shapes generated from 5 factors $F = \{$shape, scale, rotation, x and y position$\}$. To avoid label ambiguity due to the rotational symmetry of the square and ellipse shapes, we limit the range of orientations to be within $[0, \pi/2)$. Then the cardinality of each factor's space is $\{3, 6, 10, 32, 32\}$ respectively, which makes $|D| = 183,320$. MPI3D is a set of 3D datasets synthesized or recorded in a controlled environment with an object held

by a robotic arm. In our evaluation, the challenging real-world version (MPI3D-Real) is used. It has 7 factors $F$ = {object-color(6), object-shape(6), object-size(2), camera-height(3), background-color(3), horizontal-axis(40), vertical-axis(40)} with the corresponding cardinalities of the space in parentheses, leading to a total of 1,036,800 images.

## 3.2 Compositional Generalization Evaluation Protocol

Our evaluation protocol emphasizes the compositional generalization that models can achieve on downstream tasks. Our objective is to measure how *easily* an unsupervised representation learning method can produce compositional generalization using a simple model on top of the learned representation. (1) *Data splits.* We first split a dataset into train/test sets randomly while ensuring that all samples in the test-set are novel combinations of elementary factors seen in the train-set. (2) *Unsupervised representation learning.* We learn unsupervised representations from the unlabeled images of the train-set of size $N_{train}$ with a selected algorithm. (3) *Learning for downstream tasks.* Then, we freeze the learned representation model and use the $N_{label}$ labeled samples from the train set ($N_{label} << N_{train}$), to train a simple classifier / regressor to predict the ground truth value for each factor $f_i$ of the dataset. (4) *Testing generalization performance.* Lastly, we test the performance of the downstream task models on novel combinations of seen values of elementary factors.

**Readout model.** We argue that it is important to use simple read-out models and a limited number of labeled training samples to evaluate downstream tasks. Otherwise, the performance gain from the downstream task learning stage is mixed with that of the unsupervised learning stage. For example, if the unsupervised representation model is an identity mapping, we can still get great performance with a powerful read-out model and enough labeled samples. In our main article, we use linear models for downstream tasks. However, perfectly disentangled but linearly inseparable representations may still show poor performance with a linear readout model. Through sanity checking experiments (discussed in Appendix **??**), the linear readout model can still generalize well on nonlinear oracle representations possibly due to the limited value range of attributes in our datasets. In addition, extra results using a non-linear readout model (Gradient Boosting Trees) are in Appendix **??** and the main observations are consistent. The linear models we use to predict the values of the generative factors are ridge regression, using the $R^2$ score as the evaluation metric, and logistic regression, using classification accuracy as the evaluation metric. Since the $R^2$ score can be negative while $R^2 = 0$ indicates random guessing, we clip all negative $R^2$ scores to zero.

**Representation Mode.** In unsupervised disentanglement learning, low-dimensional latent variables with explicit disentanglement regularization are used as the representation for downstream tasks, e.g. the mean values of Gaussian distributions of the latent variables in the VAE. If the disentangled latent variables each represent a single ground truth factor, only a simple mapping between factors and the corresponding variables must be learned to achieve good performance. However, it is questionable if the learned latent variables disentangle in the *assumed* structure and therefore improve generalization. On the other hand, the simple linear models in our evaluation protocol may not be capable to process the latent variables (discrete messages) in emergent language (EL) learning because we would not expect discrete sequential latent variables to be easily linearly separable. Therefore, we also evaluate the intermediate features of the layers before or after the model bottleneck. Specifically, in addition to the latent variables ($z_{latent}$), we also use the features immediately after the convolutional encoder ($z_{post}$) or before the convolutional decoder ($z_{pre}$) as representations of images, shown in Fig. 1. We evaluate the use of $z_{post}$ and $z_{pre}$ in both disentanglement and emergent language models.

## 3.3 Implementation details.

**Data Splits** For the main experiments, we use a 1:9 train/test split. The train set size (10%) is smaller than what common machine learning setups and previous studies have used [36, 41]. However, considering the number of possible combinations increases exponentially for real data with an increased number of generative factors, using fewer training samples even at the unsupervised learning stage can help to obtain more meaningful conclusions for real-world scenarios.

**Model architectures.** Fig. 1 shows the architectures for disentanglement and emergent language learning models. The encoder and decoder in disentanglement learning models are symmetric architectures with a convolutional network and a multi-layer perceptron (MLP) similar to the design in [5]. We scale up the size of the model by doubling the width of all layers, which improves the

performance of all models. Since our emergent language learning model uses the same autoencoding task as disentanglement learning, it also uses a symmetric encoder-decoder architecture. Instead of MLPs, the EL model uses LSTM modules to encode and decode latent variables. The sizes of the MLP module in the disentanglement models and the LSTM module in the EL models are matched for a fair comparison. We use a Bernoulli decoder for the autoencoding task, which uses pixel values normalized to [0, 1] as probabilities.

**Hyperparameters.** We follow previous studies [31, 36] for disentanglement models to set our hyperparameters. We set the number of disentangled latent variables, $|\mathbf{z}_{latent}|$, and the maximum length of the emergent latent message, $n_{msg}$, to 10, unless specifically mentioned. For the pre-training stage, we use batch size 64 and train the model for 500,000 steps for dSprites and 1,000,000 steps for MPI3D-Real using the Adam optimizer with learning rate 0.0001. For each specific model, we collect results from three different runs with different random seeds.

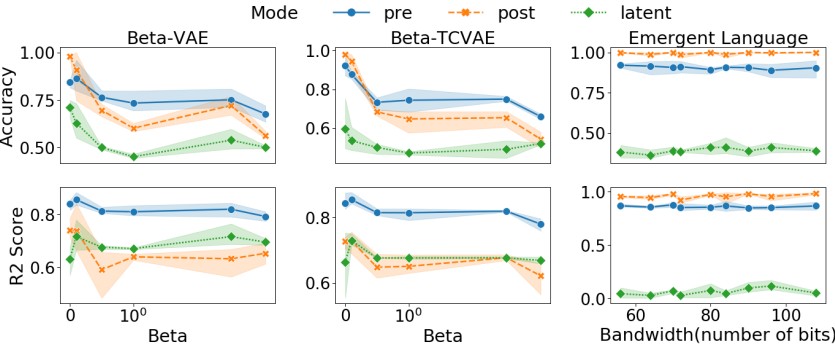

Figure 2: Generalization performance (accuracy for classification tasks and R2 score for regression tasks) of three representation models: $\beta$-VAE, $\beta$-TCVAE, and emergent language on dSprites.

# 4 Key Studies and Results

## 4.1 Compositional latent variables may not be the best representations for downstream tasks

We first study the generalization behaviors of different representation modes. For each unsupervised learning algorithm, we vary the hyperparameters that were designed to control the compositionality of representations. For $\beta$-VAE and $\beta$-TCVAE, we simply vary $\beta$. For emergent language models, we use the communication bandwidth, defined as the number of bits in the message and calculated by $\log_2(n_V^{n_{msg}})$, $n_{msg} \in \{8, 10, 12\}$ and $n_V \in \{128, 256, 512\}$. Fig. 2 shows the results for accuracy and $R^2$-score vs. $\beta$ and the number of bits when $N_{label} = 500$ for the dSprites dataset. We highlight the following observations:

**Emergent language learning** As expected, linear models do not work well on $\mathbf{z}_{\text{latent}}$ for EL models. $\mathbf{z}_{\text{post}}$ and $\mathbf{z}_{\text{pre}}$ are more suitable. $\mathbf{z}_{\text{post}}$ consistently outperforms $\mathbf{z}_{\text{pre}}$, which reveals that the emergent language bottleneck induces compositional latent structures useful for downstream tasks after being processed by the DecLSTM module.

**Disentanglement learning** Surprisingly, mismatched with the goal of disentanglement, $\mathbf{z}_{\text{latent}}$ in $\beta$-VAE and $\beta$-TCVAE is not the optimal choice for downstream tasks. For $\beta$-VAE and $\beta$-TCVAE, increasing $\beta$ to decrease the bandwidth of the bottleneck (which was thought of as the source of disentanglement [5]) *reduces* generalization performance. When $\beta$ is larger, $\mathbf{z}_{\text{pre}}$ works best. For most cases, the representation of $\beta = 0$ is best. When $\beta = 0$, the regression task seems to favor $\mathbf{z}_{\text{pre}}$ while $\mathbf{z}_{\text{post}}$ performs better for classification tasks.

**Implications.** Both disentangled representation learning and emergent language learning try to induce compositionality through the inductive bias on the representation's compositional structure. The $\mathbf{z}_{\text{latent}}$ representation itself does not perform well as its structure may be too complex to be processed by simple linear models. But if $\mathbf{z}_{\text{latent}}$ can generalize well, its decoded version $\mathbf{z}_{\text{post}}$ will perform well. In EL models, $\mathbf{z}_{\text{post}}$ consistently performs better than $\mathbf{z}_{\text{pre}}$, which demonstrates the improved generalization with the language-like bottleneck. However, disentanglement models may

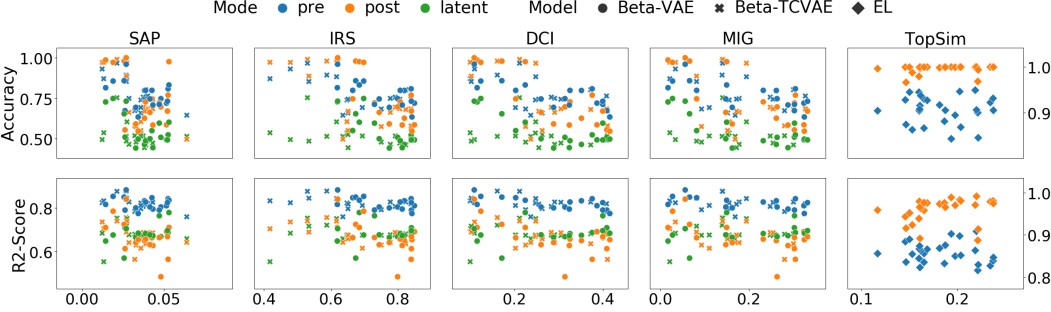

Figure 3: Compositionality metrics vs generalization performance in the dSprites dataset. The disentanglement metrics (SAP, IRS, DCI, MIG) of the $\beta$-VAE (dots) and $\beta$-TCVAE (crosses) models are not positively correlated with generalization performance in all the three representation modes. The compositionality metric for emergent language (EL), topographical similarity (TopSim), shows no strong correlation with generalization performance.

learn useless disentangled representations since increasing the penalty on either the KL-term of the ELBO or the total correlation consistently lead to worse generalization performance. To further confirm this conclusion, we also measure the disentanglement metrics on the learned $\beta$-VAE and $\beta$-TCVAE models in the next section.

## 4.2 Compositionality Metrics May Not Represent Generalization Performance

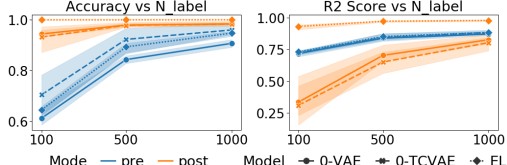

Figure 4: Generalization performance vs $N_{label}$. Three representation modes of $\beta$-VAE with $\beta$=0, $\beta$-TCVAE with $\beta$=0, and emergent language (EL) with $n_V$=256 are evaluated.

Figure 5: Generalization performance (with $N_{label} = 500$) of $\beta$-VAE with $\beta$=0, and Emergent language (EL) with $n_V$=256 on MPI3D-Real when using (5%) and (10%) unlabeled data.

We further evaluate existing compositionality metrics on learned representations. For $\beta$-VAE, $\beta$-TCVAE with various $\beta$, we measure four common disetanglement metrics: Separated Attribute Predictability (SAP) Score [27], Interventional Robustness Score (IRS) [43], Disentanglement-Completeness-Informativeness (DCI) [11], and Mutual Information Gap (MIG) [7] based on the implementation in disentanglement-lib [31]. The large-scale studies used by Locatello et al. 2019 [31] broadly measured these metrics on a few datasets. For emergent language learning models, we vary the number of bits in the discrete messages and measure topographical similarity, a common compositionality metric for emergent languages that uses the (Spearman) correlation $\rho_{Spearman}$ between the pairwise distances of the inputs and the distances of the corresponding representations. We compute the cosine distance for input attributes and the editing distance for messages. Since we have the train/test data split for the unsupervised learning stage, we evaluate these metrics on both splits. The metrics on these two splits are similar, and we only show the ones for the train split.

We find that both $\beta$-VAE and $\beta$-TCVAE show positive correlations between $\beta$ and the disentanglement metric scores, stronger in dSprites and weaker in MPI3D-Real where the highest scores usually occur for moderate $\beta$ values. However, neither of the two datasets show the same pattern as in Fig. 2 where generalization performance is negatively correlated with $\beta$. Therefore, for $\beta$-VAE and $\beta$-TCVAE, models with higher disentanglement scores do not achieve a better generalization performance. Fig. 3 confirms the noncorrelation or even negative correlation between compositionality metrics and generalization performance on the dSprites dataset. The topographical similarity of emergent language models does not show a strong correlation, e.g. $\rho_{Spearman} < 0.5$ with compositional

generalization for both dSprites and MPI3D-Real datasets. The observations are consistent with some previous studies on emergent language [6] and disentanglement [31].

**Implications.** Existing compositionality/disentanglement metrics do not measure the generalization performance of learned representations. However, we should be careful about interpreting the results as compositionality in representation learning does not necessarily help compositional generalization. As these metrics were defined more or less based on compositional annotations by humans, they can only measure particular types of compositionality. However, the compositionality exhibited in representation learning, especially with unsupervised algorithms, may not match with human definitions, e.g., a language where editing distance is a poor measure of similarity, and therefore cannot be captured by those metrics. Since designing generic compositionality is challenging or maybe even impossible, one should directly evaluate the compositional generalization if it is the ultimate goal.

### 4.3    Representations Learned by Emergent Language Models Generalize Better

In Fig 2, we observed some clues that "post" representations of EL models generalize consistently well on both tasks. In this section, we take a closer look at the generalization performance of representations learned by EL by comparing them with disentanglement models.

**Varying the number of labeled samples for downstream learning.** We evaluate the performance of downstream models trained with different numbers of labeled samples $N_{label} \in \{100, 500, 1000\}$. We compare three models: $\beta$-VAE and $\beta$-TCVAE with $\beta = 0$ (since $\beta = 0$ consistently performs best, as shown in Fig. 2) and the EL model with $n_V = 256$. Fig. 4 shows the results for dSprites. When $N_{label}$ is low, e.g. $N_{label} = 100$ and $500$ for dSprites, $z_{post}$ of EL consistently beats all other models in all representation modes for both classification (accuracy metric) and regression ($R^2$ score metric). More importantly, even when $N_{label}$ is large, 0-VAE/0-TCVAE/AE models do not have a consistent representation mode that shows similar performance as $z_{post}$ of the EL models: their $z_{post}$ is worse at regression and $z_{pre}$ is worse at classification.

**Using less unlabeled data for unsupervised learning.** Furthermore, we reduced the train-split ratio to 5% to test unsupervised learning algorithms with *less unlabeled data*. We compared the performance of the 0 -VAE and EL models on MPI3D-Real in Fig. 5. With 5% unlabeled training data, the post representation of EL models outperforms the pre/post 0-VAE model at both classfication or regression tasks and all $N_{label}$ with an even larger margin. In other words, the generalization performance of representations of 0-VAE models degrades faster with reduced unlabeled data.

**Implications.** Representations learned by emergent language models generalize well even with limited unlabeled data for unsupervised learning and labeled samples for downstream learning.

### 4.4    Ablations on Emergent Language Models

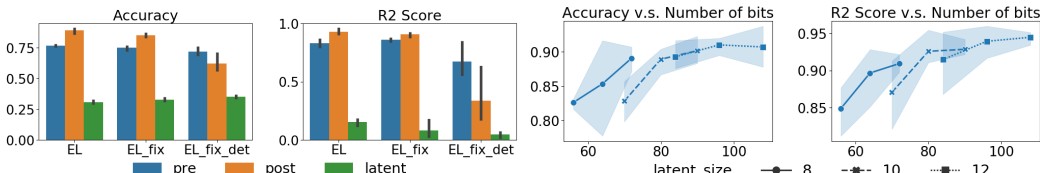

Figure 6: (Left) Ablation study of Emergent Language (EL) with $n_V = 256$ and $N_{label} = 500$ in MPI3D-Real using fixed-length messages (EL-fix) and greedy sampling (EL-fix-det). (Right) Generalization performance of EL models with $z_{post}$ and $N_{label} = 500$ on the MPI3D-Real dataset, for different bandwidths by varying message sizes $n_{msg} \in \{8, 10, 12\}$ and vocabulary sizes $n_V \in \{128, 256, 512\}$. Three $n_{msg}$ are plotted as segments of different line styles with increasing $n_V$/bits.

Since EL models showed superior compositional generalization performance over disentanglement models, we conduct ablation studies on important hyperparameters and design choices of EL models on the MPI3D-real dataset. We first study the impact of bandwidth controlled by either length of the messages or the size of dictionary. We can see that increasing the bandwidth generally improves compositional generalization. In some cases, for similar bandwidth, shorter sequences/a larger vocabulary is favored, e.g. $512^8$ vs. $128^{10}$. Since the MPI3D-Real dataset only has $6 + 6 + 2 + 3 +$

$3 + 40 + 40 = 100$ distinct values for all generative factors, a simple choice of language can express the values of factors in a fixed order using $n_{\text{msg}} = 7$ and $n_V = 100$. Apparently this behavior is not learned by the current design of EL models, possibly because of optimization issues due to poor gradient estimation for discretization, regardless of the good generalization performance. However, EL models can learn their own language with additional redundancies that require more bandwidth.

We also ablate the two characteristics of the EL models we have considered thus far: allowing variable-length messages and using stochastic sampling. We can turn an EL model into one with fixed-length messages by always generating $n_{msg}$ tokens (EL-fix) and additionally with deterministic messages by using greedy sampling (EL-fix-det). We can see that as we remove these two designs, the generalization performance is worse, especially after making the model deterministic. Notably, the EL model with fixed-length messages and using deterministic sampling (EL-fix-det) is very close to the popular discrete representation learning model VQ-VAE [46] that uses a shared vocabulary over spatial locations of the feature map. Therefore, our study may indicate that the use of discrete latent variables of variable length and stochastic sampling can enable unsupervised learning models to learn better representations for downstream tasks. We also found that while the EL model allows variable-length codes, at convergence it still almost always uses the maximum message length on both the training and testing set. This is not surprising given that the reconstruction task drives the discrete message to be longer so that more information can pass through the discrete bottleneck.

## 5    Limitations of Our Study

Our goal is to provide a comprehensive study of learning algorithms, including their hyperparameters. However, our study is limited on the variety of other design choices to restrict the experimental complexity. While we studied both synthetic and realistic image datasets, both these datasets are relatively simple with the same small number of generative factors and each of the factor follows a uniform distribution. For learning algorithms, we focus on studying the inductive bias on the representation format while fixing the model architecture design which can impact the results. Moreover, we did not study hyperparameters beyond those related to the latent representations. Specifically, we did not study how the type and configurations of the optimizer and the batch size would change the results; instead, we followed common setups in previous studies.

## 6    Related Work

Compositional generalization was shown in previous work on disentangled representation learning [12, 21] to generate images of novel combinations of concepts. Zhao et al. [50] systematically evaluated generative models in a few compositional generalization tasks without exploring whether disentanglement is correlated with generalization performance. Recently, Montero et al. [36] studied the compositional generalization along with the interpolation and extrapolation generalization of two VAE-based disentanglement models. Similar to [50], the performance is evaluated on the unsupervised learning task, e.g. image reconstruction/generation. However, we directly evaluate the compositional generalization. Different from [36] that manually selects the train-test splits with various difficulty, we use random splits which is earlier for studying different split ratios.

The comprehensive study on the unsupervised learning of disentangled representation by Locatello et al. [31] included an evaluation on downstream tasks. However, they train unsupervised models on the whole dataset and therefore are not able to evaluate the generalization as we do. [10] evaluated the downstream out-of-distribution (OOD) generalization that includes the interpolation and extrapolation generalization of unsupervised and weakly supervised disentanglement models, while we focus on compositional generalization. A very recent work [41] included compositional generalization behaviors of unsupervised and weakly supervised disentanglement models in their broad study on generalization. Unlike our evaluation that trains simple downstream models with a small amount of labeled samples, they use all labels in the train set to learn MLP models. Furthermore, none of these works evaluated the representations beyond the latent variables at the bottleneck layer as we did.

In emerging language (EL) learning, Chaabouni et al. [6] found that regardless of the degree of compositionality measured by topographic similarity, EL can generalize to novel combinations of concepts when the input space is rich enough. Andreas [1] proposed a new compositionality metric,

tree reconstruction error (TRE), measuring how well a representation model can be approximated by a compositional operator and learnable primitive representations. While claimed as a general compositionality metric with learnable compositional operator, some pre-commitment to a restricted composition function is essentially inevitable. Similar to [6], [1] also studied the relationship between TRE and generalization. While they work on simple attribute data and evaluate on the tasks for emergent language learning, we use image data and are motivated by learning useful representations for downstream tasks. Furthermore, unlike all previous work, we studied the compositional generalization of representations from both disentanglement and emergent language models with a unified setup.

## 7    Conclusions and Discussions

We proposed a protocol to evaluate the compositional generalization of unsupervised representation learning models that have a built-in inductive bias for compositionality: disentanglement models and emergent language (EL) learning. Our evaluation emphasizes using a small number of labeled samples to train simple models for downstream tasks. The interesting finding that latent variables at the bottleneck do not work as well as "pre" and "post" representations reminds us to be careful when concluding that a model performs poorly when its latent representation does not perform well. For disentanglement models, we observe that generalization performance is not well correlated with existing disentanglement metrics. This finding demonstrates the gap between pursuing better disentanglement metrics and more generalizable representations in previous studies. Similarly, the existing compositionality metrics for EL, e.g. topographical similarity, cannot represent the generalization of learned representations. However, under the same setup, EL models, which were not initially proposed for unsupervised representation learning, induce representations with surprisingly strong compositional generalization and are robust to the change of dataset and hyperparameters.

We focus on unsupervised learning algorithms that are specifically designed for compositional representation to answer whether they lead to better generalization (which was an assumption made by a great deal of prior work). A broader evaluation on other representation learning approaches e.g. self-supervised learning [8, 18] would be interesting for future work. We hope that our study draws more attention to the study of compositional generalization and emergent language models as a way of learning representations. Interesting future directions include working on more challenging datasets, e.g. images of multiple objects; testing on other downstream tasks, e.g. visual question answering; testing different model architectures, e.g. Transformers [47]; and developing better pre-training tasks beyond auto-encoding. Recent work showing that emergent language ($z_{latent}$) can be used as a corpus to pretrain a language model [48] also suggests that emergent language may be a better visual representation choice in vision-language models.

## Acknowledgement

We thank Nikhil Kandpal for giving feedback on the draft of the paper. Zhenlin Xu was supported by the Royster Society of Fellows Dissertation Completion Fellowship during this work.

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
