Supplementary Material for
# Compositional Generalization in Unsupervised Representation Learning:
# From Disentanglement to Emergent Language

**Zhenlin Xu      Marc Niethammer      Colin Raffel**
Department of Computer Science
University of North Carolina at Chapel Hill
{zhenlinx, mn, craffel}@cs.unc.edu

## A   More Implementation Details

**Model architectures.**   In our experiments, disentanglement models and emergent language (EL) models use the same architectures for the convolutional encoder and decoder. Disentanglement models use an MLP to encode the convolutional features into the disentangled latent variables and another MLP to decode them, while EL models use two LSTMs to encode and decode the discrete messages. Table 1 provides architecture details for all these modules.

Table 1: The encoder module architectures which are the symmetric reflection of the decoder layers.

| EncConv | EncMLP | EncLSTM |
|---|---|---|
| 4x4 Conv, 64 ReLU, stride 2 | FC 512, ReLU | |
| 4x4 Conv, 128 ReLU, stride 2 | FC 1024, ReLU | LSTM 512 |
| 4x4 Conv, 128 ReLU, stride 2 | FC 1024, ReLU | Linear $n_{msg}$ |
| 4x4 Conv, 128 ReLU, stride 2 | FC 512, ReLU | |
| Flatten layer | Linear $|z_{latent}|$ | |

**Readout Models.**   We use the implementations in scikit-learn[1] (version 0.22) for the readout models of our downstream evaluation. The specific linear or gradient boosting tree (GBT) models for classification or regression tasks are configured as specified in Table 2.

Table 2: The implementation details of the readout models.

| | Scikit-learn function | Configuration |
|---|---|---|
| Linear | linear_model.LogisticRegressionCV | default |
| | linear_model.RidgeCV | alphas=[0, 0.01, 0.1, 1.0, 10] |
| GBT | ensemble.GradientBoostingClassifier | default |
| | ensemble.GradientBoostingRegressor | default |

**Computational Cost.**   We use an Nvidia RTX A6000 to benchmark the computational cost. For unsupervised representation learning, training a $\beta$-VAE or a $\beta$-TCVAE model takes about 3 hours and 10 hours for dSprites and MPI3D-real respectively, and an EL model with $n_{msg} = 10$ takes about 11 and 15 hours respectively for dSprites and MPI3D-real .

---

[1] https://scikit-learn.org/

36th Conference on Neural Information Processing Systems (NeurIPS 2022).

Table 3: Sanity check with oracle linear and non-linear representations. The accuracy/R2-score are reported for classfification and regression tasks respectively.

| Representations | Linear | GBT |
|---|---|---|
| attributes | 100% /100% | 100% / 100% |
| attributes$^2$ | 94.7% / 100% | 100% / 100% |

**Dataset License.** The two datasets used in our experiments, dSprites[2] and MPI3D[3], are publicly available under an Apache License and the Creative Commons Public License respectively.

**Reproducibility.** Our code is publicly available.[4]

## B  Sanity Check Experiments with Oracle Representations

We test the oracle representations using the ground truth value of all attributes or the squared value of each attribute (which would not be perfectly linearly fittable). On the dSprites dataset, we use 500 samples to train linear or GBT readout models for classification and regressions tasks. Their generalization performance is given in Table 3. The attribute values are expected to generalize perfectly with linear or GBT readout models. However, linear readout models can still fit the non-linear attributes[2] well. We think this may be due to the limited value range of attributes of our datasets. This experiment shows that if the learned representation is disentangled into attributes (as is often the goal), the linear head should not be a major issue to constrain the generalization performance.

## C  Detailed Experimental Results

In the main article, we only present representative results to support our key findings. In this section, we provide detailed results for different algorithms and datasets with additional gradient-boosting-tree (GBT) read-out models that are able to model non-linear mappings.

**Compositional latent variables may not be the best representations for downstream tasks.**
Fig. 1 compares $pre$, $latent$, and $post$ representation modes for three different learning models ($\beta$-VAE, $\beta$-TCVAE, and emergent language (EL)) using different read-out models. For emergent language (EL) models, when using GBT instead of linear models for downstream tasks, $\mathbf{z}_{latent}$ performance improves, but still underperforms $\mathbf{z}_{pre}$ and $\mathbf{z}_{post}$. For disentanglement models, increasing the regularization ($\beta$) still decreases performance when using GBT read-out models. However, when $\beta = 0$, the regression task no longer favors $\mathbf{z}_{pre}$ and $\mathbf{z}_{post}$ performs well for both the regression and classification tasks.

**Compositionality Metrics May Not Represent Generalization Performance.** Fig. 2 shows the disentanglement/compositionality metrics vs generalization performance on the dSprites and MPI3D-Real datasets using linear and GBT read-out models. Consistently to our results in the main article, we do not observe strong correlations between these metrics and generalization performance. Figs. 3 and 4 show quantitative measures of ranking correlation. We see that for disentanglement models, all metrics show no or negative correlations with generalization performance except for a weak correlation between the DCI score and generalization on the MPI3D-Real dataset. For EL models, $post$ representations show stronger, although still weak, correlations than $pre$ representations.

**Representations Learned by Emergent Language Models Generalize Better.** Fig. 5 compares EL models with $\beta$-VAE and $\beta$-TCVAE with $\beta = 0$. We see that $\mathbf{z}_{pre}$ of EL using linear read-out models gives the best performance overall especially when $N_{label}$ is small. While applying GBT read-out models improves the performance of $\mathbf{z}_{post}/\mathbf{z}_{pre}$ over the 0-VAE and 0-TCVAE models and $\mathbf{z}_{pre}$ from the EL models in regression tasks, GBT read-out reduces performance in other cases,

---

[2]https://github.com/deepmind/dsprites-dataset
[3]https://github.com/rr-learning/disentanglement_dataset
[4]https://github.com/wildphoton/Compositional-Generalization

especially for classification tasks. The reason may be that GBT models need more labeled samples than linear models for training to work well. When we reduce the train-split ratio in Fig. 6, the EL model learns more generalizable representations than the 0-VAE and 0-TCVAE models which degrade faster with reduced unlabeled data.

**Ablations on Emergent Language Models.** From the ablation study of EL models in Figs. 8 and 7, we observe consistent patterns: using a shorter sequence with a larger vocabulary size works better; using greedy sampling significantly reduces the performance of EL models.

**A closer look of generalization.** Our evaluation protocol includes two training stages: unsupervised pre-training of a representation model and supervised training of a read-out model with a small amount of labeled data. To better understand the generalization results, we further evaluate performance on the whole unsupervised training data (US-train) that is only visible to the representation model, and the supervised training data (S-train) that is part of the pre-training data and were seen by both the representation model and the readout model. The results on the held-out test set (Test) unseen by both training stages are given as a reference. In Table 4, we show results of evaluating the pre/latent/post representations of EL and $\beta$-VAE(beta=0) with linear or GBT readout models on the dSprites dataset. The classification accuracy or regression R2 score is given in each entry.

- On all representation models/modes and read-out models, the performance of Unsup-train and test is very close. This tells us that an example seen by the unusupervised pretraining stage does not necessarily have good performance in a downstream task in our setting.

- In the S-train set, linear readout models under-fit the latent representations of both VAE and EL models. Bad performance is expected for EL-latent representations since a linear readout model is not a good choice for language-like messages as discussed in the paper. Combined with sanity-checking experiment B, it indicates that VAE models do not produce representations that disentangle attributes.

- GBT readout models fit EL-latent and VAE-latent well on S-train but generalize poorly to US-train and Test. This further indicates that latent representations perform worse in providing good generalization when compared to pre/post representations.

Table 4: Performance on three subsets of dSprites data. Pre/latent/post representations of EL and $\beta$-VAE(beta=0) are evaluated with linear and GBT readout models. The classification accuracy / regression R2 score is given in each entry.

| Data + Readout | VAE-Pre | VAE-Latent | VAE-Post | EL-Pre | EL-Latent | EL-Post |
|---|---|---|---|---|---|---|
| S-train + Linear | 99.93/94.48 | 73.27/64.83 | 100/95.99 | 100/92.55 | 46.13/13.35 | 100/99.39 |
| US-train + Linear | 84.83/84.15 | 70.89/63.13 | 98.74/71.29 | 91.02/84.7 | 39.11/9.92 | 99.99/98.04 |
| Test + Linear | 84.3/83.9 | 70.98/63.1 | 97.88/73.98 | 90.56/84.6 | 38.8/9.5 | 99.94/97.85 |
| S-train + GBT | 100/96.44 | 99.33/93.64 | 100/98.81 | 100/95.86 | 96.93/83.63 | 100/99.83 |
| US-train + GBT | 77.67/79.44 | 76.8/77.57 | 97.73/88.34 | 81.58/81.02 | 56.19/58.53 | 99.53/95.83 |
| Test + GBT | 76.76/78.95 | 76.09/77.26 | 96.83/87.82 | 80.86/90.56 | 54.84/57.36 | 99.52/95.68 |

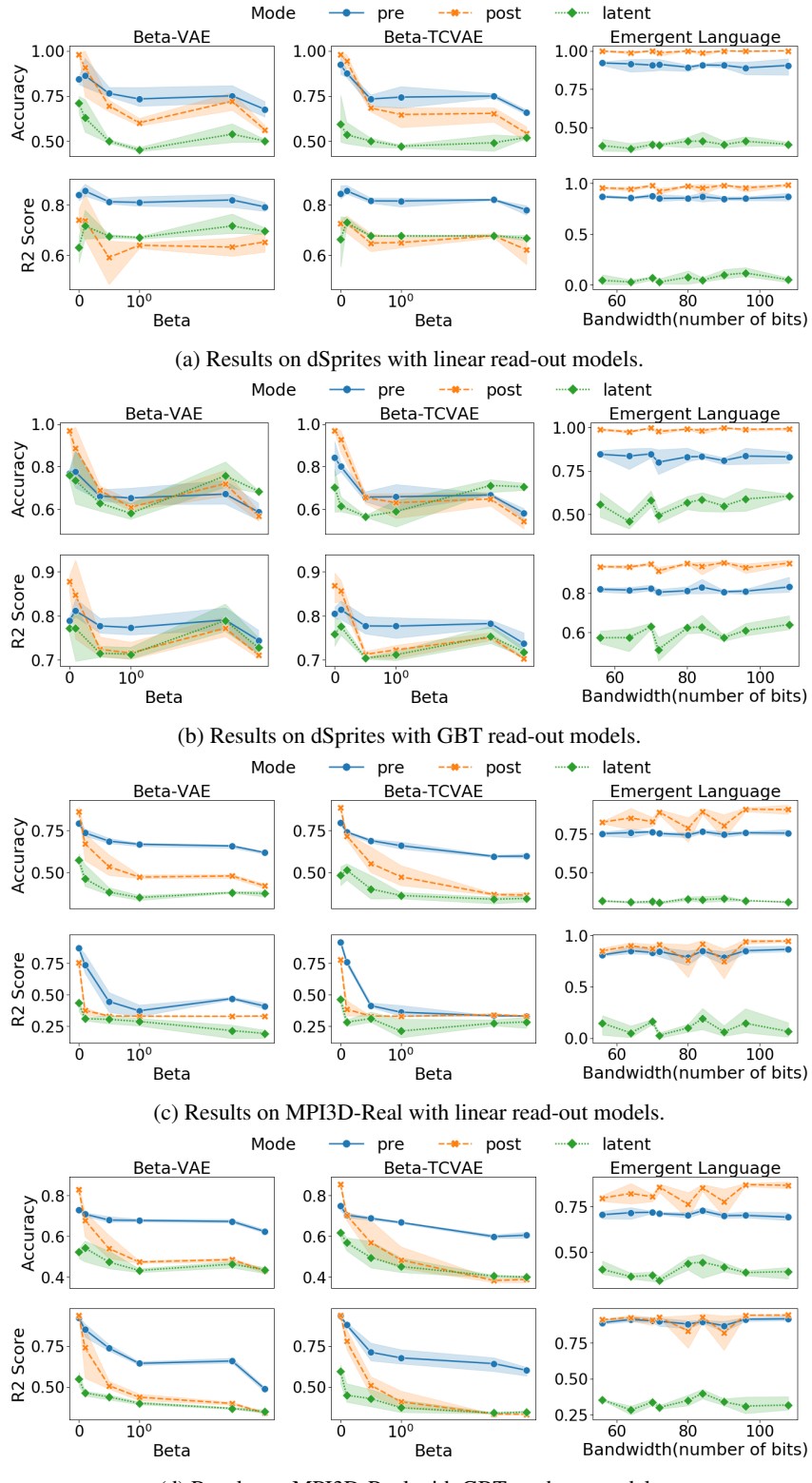

(a) Results on dSprites with linear read-out models.

(b) Results on dSprites with GBT read-out models.

(c) Results on MPI3D-Real with linear read-out models.

(d) Results on MPI3D-Real with GBT read-out models

Figure 1: Generalization performance (accuracy for classification and R2 score for regression) with $N_{label} = 500$ of three representation models: $\beta$-VAE, $\beta$-TCVAE, and emergent language (EL) varying hyper-parameters ($\beta$ or bandwidth), datasets (dSprites and MPI3D-Real) and read-out model (linear and GPT).

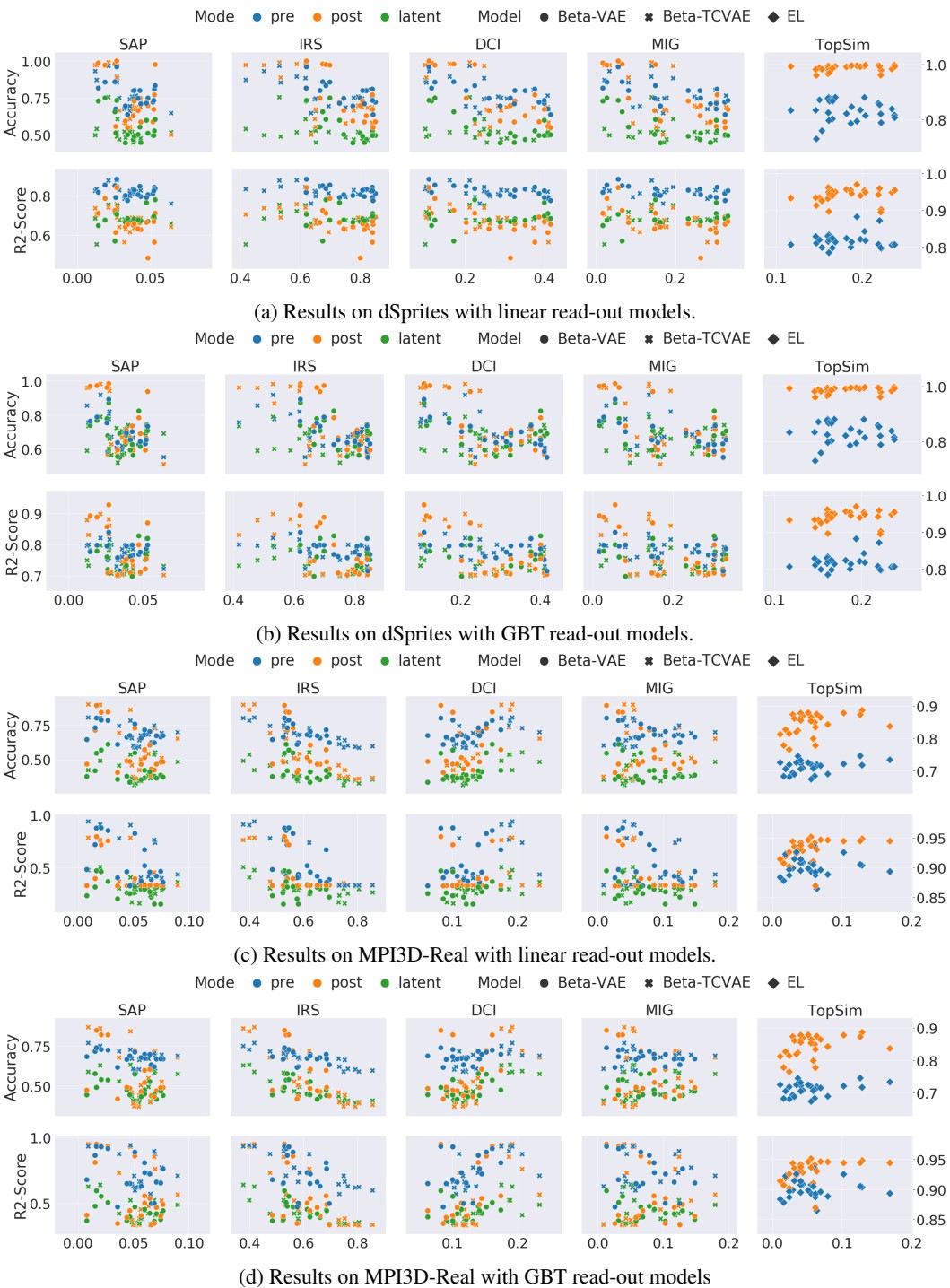

(a) Results on dSprites with linear read-out models.

(b) Results on dSprites with GBT read-out models.

(c) Results on MPI3D-Real with linear read-out models.

(d) Results on MPI3D-Real with GBT read-out models

Figure 2: Compositionality metrics vs generalization performance on dSprites and MPI3D-Real datasets, and linear and GPT read-out models. The disentanglement metrics (SAP, IRS, DCI, MIG) of the $\beta$-VAE (dots) and $\beta$-TCVAE (crosses) models are not positively correlated with generalization performance in all the three representation modes. The compositionality metric for emergent language (EL), topographical similarity (TopSim), shows no strong correlation with generalization performance.

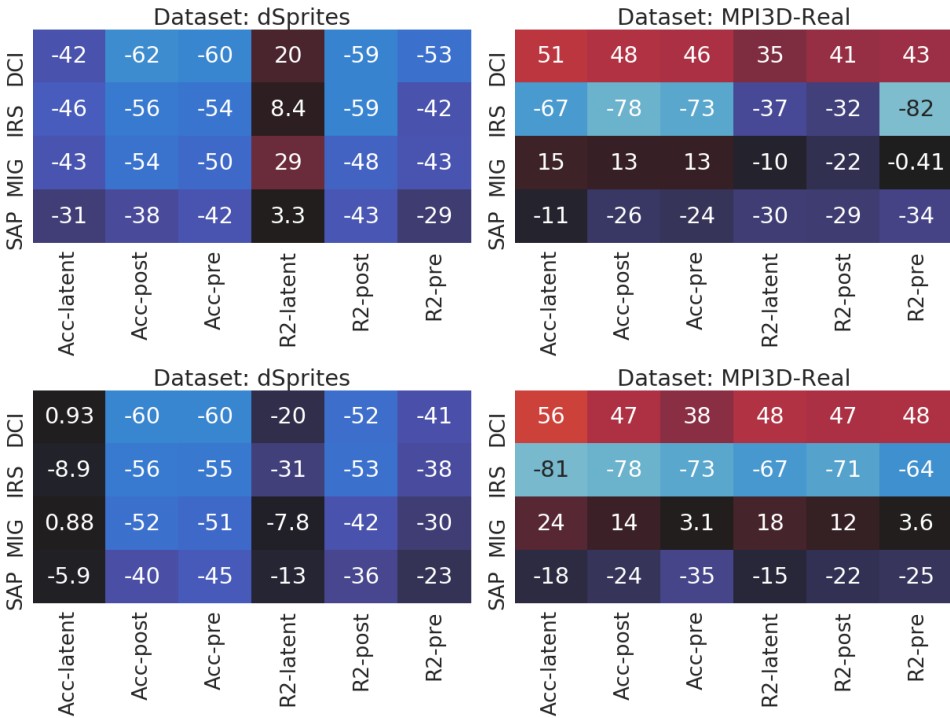

Figure 3: Ranking correlation between disentanglement scores (SAP, IRS, DCI, MIG) and the generalization performance of three representation modes (pre, latent, post) using linear (the first row) and GBT (the second row) read-out models on dSprites (the left column) and MPI3D-Real (the right column) datasets. Except for the DCI metric, which shows weak correlations with generalization performance on MPI3D-Real, all other metrics show no or even negative correlations.

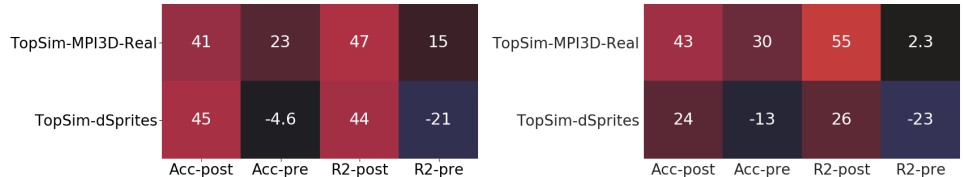

Figure 4: Ranking correlation between topographical similarity (TopSim) and the generalization performance of three representation modes (pre, latent, post) using linear (the left column) and GBT (the right column) read-out models on dSprites (the second row) and MPI3D-Real (the first row) datasets. The correlations between TopSim and generalization are stronger on *post* representations than on the *pre* representations.

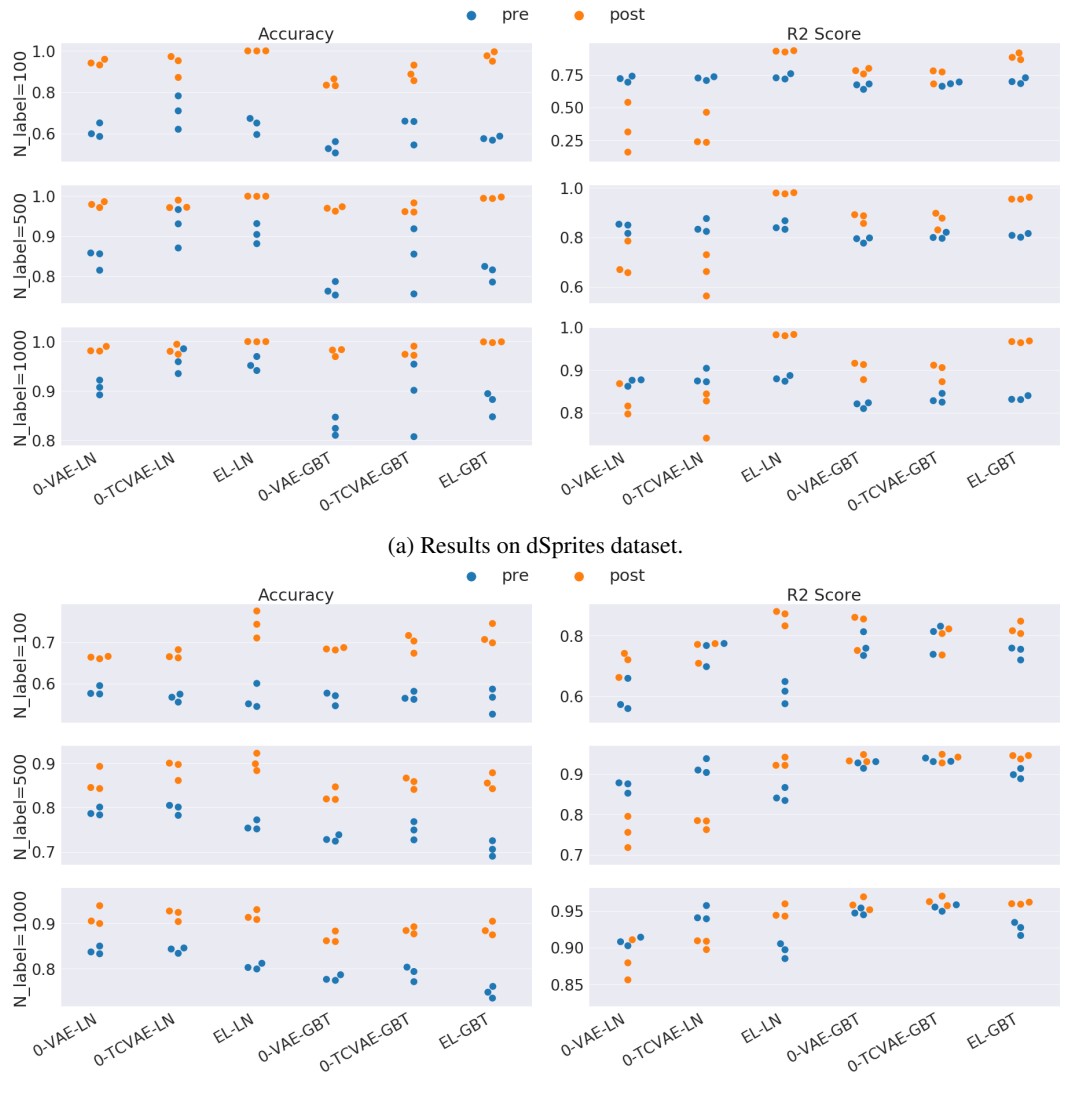

(a) Results on dSprites dataset.

(b) Results on MPI3D-Real dataset.

Figure 5: Generalization performance of two representation modes (pre, post) of $\beta$-VAE with $\beta$=0, $\beta$-TCVAE with $\beta$=0, and emergent language (EL) with $n_V$=512 when evaluated with linear (LN) and gradient boosting tree (GBT) read-out models.

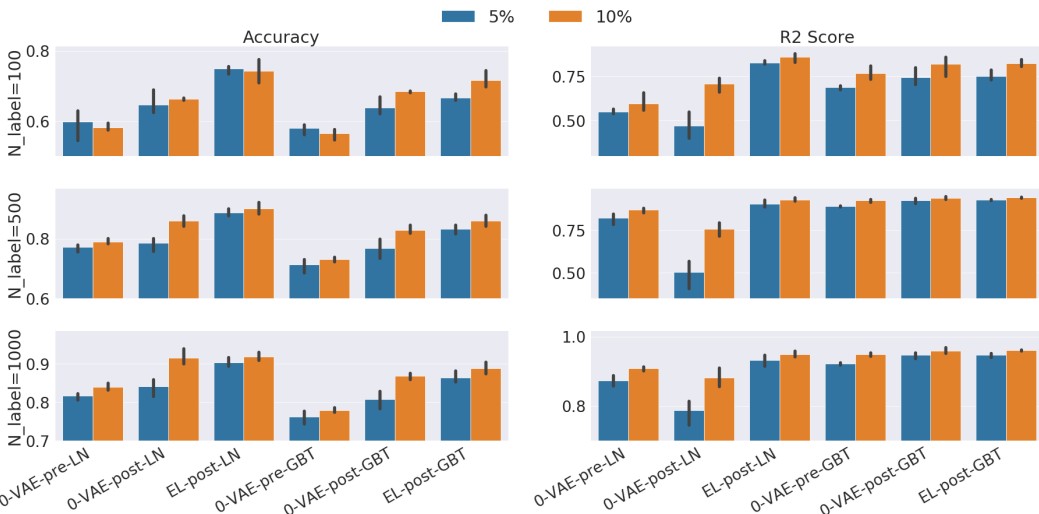

Figure 6: Generalization performance of *pre* and *post* representations of $\beta$-VAE with $\beta$=0 (0-$\beta$-VAE) and $\beta$-TCVAE with $\beta$=0 (0-$\beta$-TCVAE), and *post* representations of emergent language (EL) with $n_V$=512 when evaluated with linear (LN) and gradient boosting tree (GBT) read-out models on MPI3D-Real when using (5%) and (10%) unlabeled data.

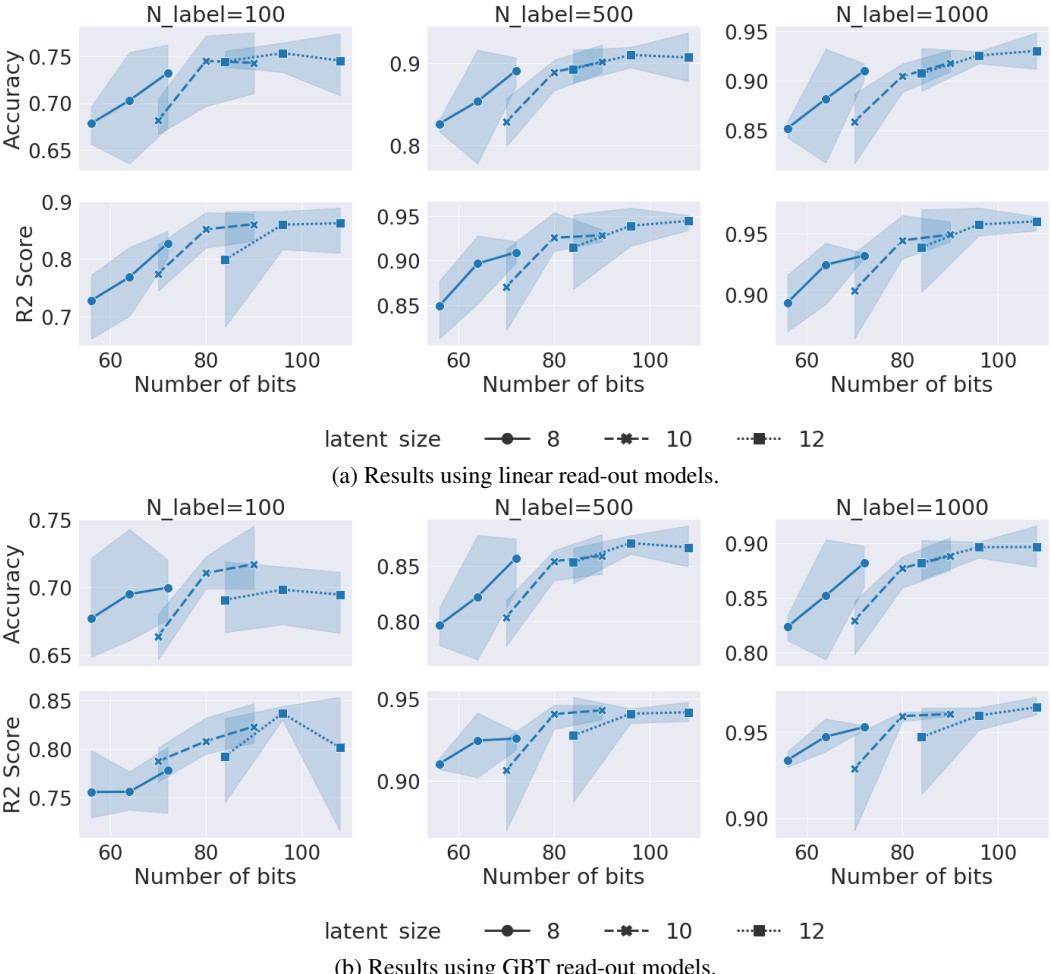

Figure 7: Ablation study of Emergent Language (EL) models with $z_{post}$ and $N_{label} = 100/500/1000$ on MPI3D-Real dataset evaluated with linear (a) and gradient boosting tree (b) read-out models, for different bandwidths by varying message sizes $n_{msg} \in \{8, 10, 12\}$ and vocabulary sizes $n_V \in \{128, 256, 512\}$. The three $n_{msg}$ results are plotted as segments of different line styles with increasing $n_V$/bits.

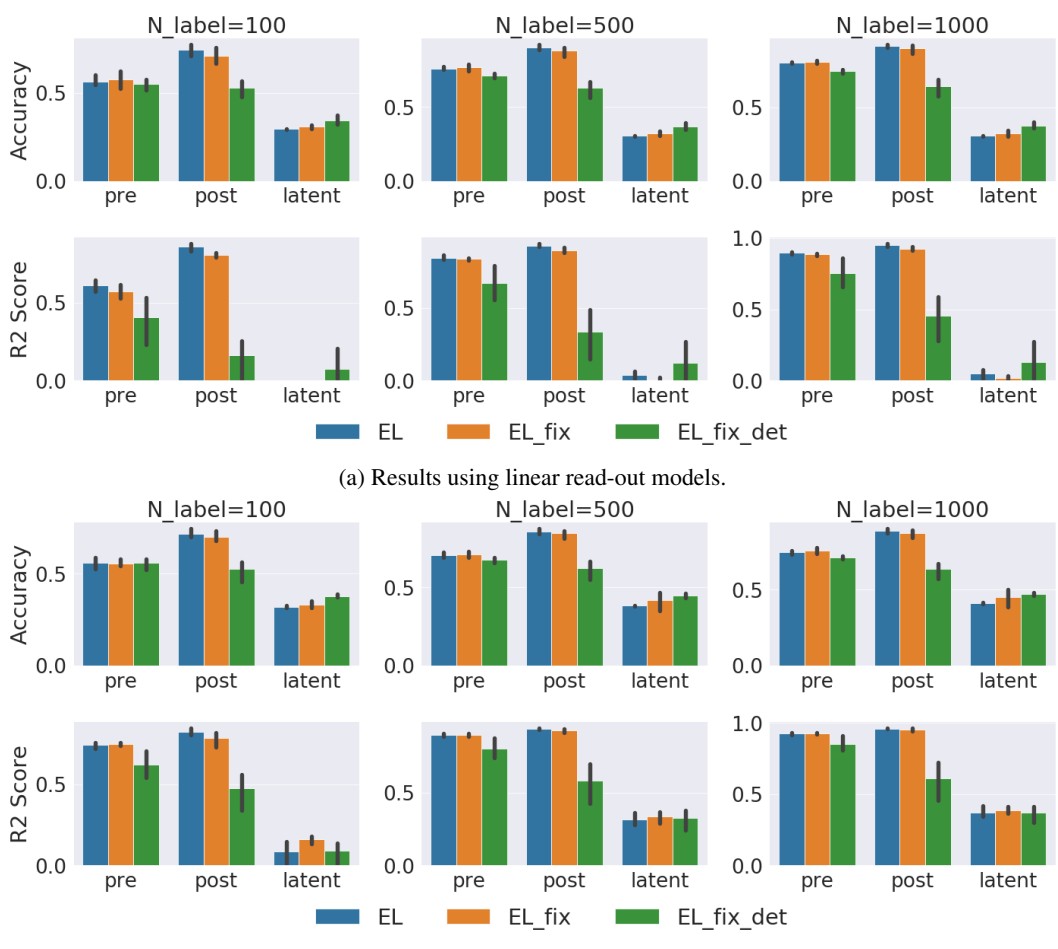

(a) Results using linear read-out models.

(b) Results using GBT read-out models.

Figure 8: Ablation study of Emergent Language (EL) with $n_V = 512$ and $N_{label} = 100/500/1000$ for MPI3D-Real when using fixed-length messages (EL-fix) and greedy sampling (EL-fix-det).