# OpenReview forum: "Compositional Generalization in Unsupervised Compositional Representation Learning: A Study on Disentanglement and Emergent Language"
_NeurIPS.cc/2022/Conference — NeurIPS 2022 Accept_

### Official Review · Reviewer_mvHF · 2022-07-08

**Rating:** 6
**Confidence:** 4
**Soundness:** 3 good
**Presentation:** 4 excellent
**Contribution:** 3 good

**Summary:**

The authors study the compositional generalization performance of several approaches which aim to learn compositional representations including those focused on disentanglement and those based on emergent language. They propose a new metric to measure compositional generalization, specifically exploring novel combinations of previously-observed concepts. They find that, perhaps counterintuitively, often the features immediately before or after the latent variable perform better on generalization metrics. They evaluate on multiple instances of each model and also analyze the relationship between the generalization performance metrics and ones commonly used to evaluate disentanglement

**Questions:**

“we find that the previously proposed metrics for evaluating the levels of compositionality are not correlated with actual compositional generalization in our framework” - Why might this be? The paragraph on Implications in Sec 4.2 discusses certain conclusions which cannot necessarily be drawn, but what does it imply if “compositionality exhibited in representations learning …  may not match with [the] human definition”? If they are different, how might they be different?

**Limitations:**

The limitations section in the Appendix is reasonable and complete.

**Strengths And Weaknesses:**

Strengths:
- The results are non-obvious and have implications for other representation learning research which, for autoencoders, often (though not always) focuses evaluations on the latent representations.
- The experiments are thorough, covering a broad range of interesting questions, and well-grounded in disentanglement literature. In particular, this sheds some important light on questions left open by the widely-referenced Locatello et al. 2018., as discussed in Section 5.
- The metrics are reasonable, the datasets are standard for the disentanglement literature (though more would always be welcome), and the models are also reasonably chosen.

Weaknesses:
- Given the paper’s title and claims, other kinds of representation-learning approaches would have been reasonable (e.g. encoder-only self-supervised techniques like SimCLR or BYOL, masked autoencoders, decoder-only ones like InfoGAN coupled with projection). However, the results carry useful implications even without this.

---

> ### Author Response · Authors · 2022-08-02
> **Response to Reviewer mvHF**
>
> Thank you very much for your feedback and suggestions. Below are our response to your comments.
>
> *Suggestion 1: include representation-learning approaches e.g. encoder-only self-supervised techniques like SimCLR or BYOL, masked autoencoders, decoder-only ones like InfoGAN coupled with projection.*
>
> - We agree that evaluating the compositional generalization of mentioned representation learning approaches would be interesting for future work. We want to highlight that we focus on unsupervised representation learning approaches that are specifically designed for compositional representation learned and we aim to answer whether they lead to better generalization (which was an assumption made by a great deal of prior work).
> - On the other hand, evaluating the suggested representation learning methods on dSprites/MPI3D may be problematic as they more or less include/require some assumptions on the data properties besides compositionality. For example, contrastive learning methods like SimCLR/BYOL require significant data augmentation. Augmentation techniques like cropping and color jittering will change the attributes/representation in the datasets we study. MAE also assumes correlations between local image patches which is not a valid assumption in our single object data.
>
> *Question: What is the possible reason for our finding that “the previously proposed metrics for evaluating the levels of compositionality are not correlated with actual compositional generalization in our framework”. How the “compositionality exhibited in representations learning” be different with “[the] human definition”.*
>
> The poor correlation could be understood in different ways:
> - If the compositionality metrics work perfectly, the poor correlation indicates that the compositionality of representations does not necessarily lead to generalization. However, we should be careful about taking this conclusion as the metrics may have flaws.
> The model may not learn *desired* compositionality. Due to the unsupervised nature of studied learning algorithms, there is no guarantee that learned representations will behave as we want them to. We validate this insofar as we are able to rely on specific existing compositionality metrics.
> - Maybe some compositionality is emergent but misaligned with what is defined in our metric. For example, the unsupervised disentanglement learning model applies regularization on dependence/correlation latent variables. However, this does not guarantee the emergence of attributes, measured in disentanglement metrics. A straightforward example is, to represent the 2D locations, any rotated version of the ground truth coordinate system would be a valid disentangled representation although it does not match with the metric based on human-defined attributes. Another example is that an EL model learns a language for which edit distance is not a good similarity metric and therefore topographical similarity cannot capture its compositionality.

---

> > ### Comment · Reviewer_mvHF · 2022-08-07
> > **Thanks for the response**
> >
> > Thank you for the response! I think the discussion of compositionality vs compositional generalization is useful.
> >
> > I think that, at present, the title and abstract somewhat overstate the generality of the results given they specifically focus on a relatively narrow set of representation learning techniques (even within VAE-based approaches are not analyzed, e.g. FactorVAE). The way that compositional generalization is defined and explored in this paper does not strictly require that the methods learn representations where the latent dimensions themselves are disentangled (or else, one might as well look at disentanglement metrics directly) - in fact, as stated in the abstract, "increasing pressure to produce a disentangled representation produces representations with worse generalization." Given this, there is no reason to believe that representation learning approaches which are not explicitly trained for disentanglement would be worse at compositional generalization (in fact, given this claim, it would not be unreasonable if they were better). Moreover, it appears that one can artificially lower the disentanglement performance (in most metrics that depend on disentangled latent factors) of these models with an affine transform without affecting the proposed compositional generalization metric.
> >
> > To be clear, the results presented are still insightful and valuable, and the framework presented is clearly a powerful tool for an improved understanding of representation learning methods. Still, there are many other representation learning techniques that may be capable of compositional generalization which have not been analyzed in this paper. Considering all this, I am keeping my score as it is.

---

> > > ### Author Response · Authors · 2022-08-08
> > > **Re: Thanks for the response**
> > >
> > > Hi Reviewer mvHF, thanks very much for responding. We appreciate the point that other representation learning methods that do not explicitly target disentanglement or compositional generalization could perform well in our evaluation framework. Given that considering the full spectrum of representation learning techniques is out of the scope of our paper and computational budget, we are wondering if there is any way we could change the title and/or abstract to address your concern and make your take on the paper more positive overall.

---

### Official Review · Reviewer_tbrQ · 2022-07-11

**Rating:** 5
**Confidence:** 2
**Soundness:** 3 good
**Presentation:** 3 good
**Contribution:** 2 fair

**Summary:**

The paper aims to investigate the compositional generalization performance of the representation learned from unsupervised learning, focusing on disentanglement and emergent language (EL). In contrast to previous works that measure compositional generalization on the learning task, the authors propose a two-stage evaluation protocol that does the same but on downstream tasks by introducing a simple header on top of the learned representation. Through experiments on two image tasks, they find that (1) representations from layers before or after the model bottleneck result in better compositional generalization (4.1); (2) previously proposed compositionality or disentanglement metrics do not always correlate with better performance (4.2); (3) EL appears to be more powerful than disentanglement models in such generalization (4.3). An ablation study follows to confirm the function of EL (e.g., hyperparameters).

**Questions:**

1). In addition to the explored factors on lines 88-90, varying the model structure as the backbone of both encoder and decoder is needed.

2). Can EL predicts correctly for failed cases of disentanglement models, or the opposite?

3). Is there a case that leads to higher compositionality metrics but lower compositionality?

4). Other than dSprites and MPI3D-Real, I'm curious about whether the same findings can be observed from any application tasks.


**Limitations:**

I did not see a discussion regarding the limitations.

**Strengths And Weaknesses:**

Strengths:1). This paper focuses on the valuable problem of systematic generalization (i.e., generalization from learned concepts to unseen compositions) that plays a crucial role in improving the generalization of models and the following interpretation. 2). The work is well-motivated and focuses on compositional representation learning. The introduced evaluation protocol sounds reasonable and may be helpful for the followers to refer to. 3). The experimental results on two benchmarks are encouraging and supported by a further ablation study. The findings may benefit the usage of learned representations and distinguish the advance of EL over disentanglement models in compositional generalization.

Weaknesses:1). Technically speaking, the contribution of this work is incremental. As an analysis paper, its technique depth is shallow. 2). The reported experimental results appear to evidence the findings, while the encoder and decoder are fixed. More model structures. (e.g., transformers) should be taken into consideration to support the findings. 3). The work ends with the ablation study on EL; however, more studies and case analysis are necessary to underline the difference between the two learning methods (e.g., a failed case of disentanglement models but correctly inferred by EL).

---

> ### Author Response · Authors · 2022-08-02
> **Response to Reviewer tbrQ**
>
> We appreciate your feedback and the recognition of our motivation and the importance of our findings. Your concerns and questions are addressed below.
>
> *Concerns #1: As an analysis paper, its technique depth is shallow.*
>
> Our work focuses on revealing behaviors of existing models on the compositional generalization that were missed in previous studies on disentanglement/compositionality. Although we did not build new learning algorithms or model architectures, we believe that designing a sensible evaluation protocol and implementing and evaluating existing algorithms under a fair and unified framework constitutes a significant contribution.
>
> *Concern #2 and Question #1: More model structures. (e.g., transformers) should be taken into consideration to support the findings.*
>
> We agree that studying more model architectures including transformers would be interesting future work. Given the limited timeline, we are not able to do them during rebuttal. In this paper, we focus on the evaluation and analysis of previously proposed designs that were widely validated. As we discussed in section 3.3, the model architecture we use were designed by simply scaling up the model in previous disentanglement work. Using similar designs enables easier connections between our findings with previous research. In our preliminary experiments, we observed consistent behaviors on the model without scaling up.
>
> *Concern #3 and Question #2: More analysis on the difference between the two learning methods (e.g., a failed case of disentanglement models but correctly inferred by EL).*
>
> We appreciate the suggestion of more analysis. It is challenging to draw general conclusions by looking at individual examples where two algorithms behave differently. However, we do observe that beta-VAE works significantly worse than EL models on specific attributes e.g. orientation of dSprites dataset. We will include attributes-wise results in supplementary material in our updated version.
> Question #3:  Is there a case that leads to higher compositionality metrics but lower compositionality?
> No. Compositionality can only be quantified by the compositionality metrics up to the choice of concrete format of compositionality. Therefore, a higher compositionality metric means higher compositionality although specific to the kind the chosen metric is designed to measure.
>
> *Question #4: Other than dSprites and MPI3D-Real, I'm curious about whether the same findings can be observed from any application tasks.*
>
> Evaluation of other downstream tasks is definitely an interesting direction for future work. Due to the page limits, we will leave it as future work. We do want to note that it is challenging to control the compositionality of novel test set as most real applications don’t have the structures of samples labeled.
>
> *Concern #4: missing discussion regarding the limitation.*
> We have a limitation discussion in section B of supplementary material.

---

> > ### Comment · Reviewer_tbrQ · 2022-08-07
> > **Response to authors**
> >
> > Thanks for the response. Most of my concerns have been addressed, and I have found more details in the supplementary. I hope to see this paper at the conference, while I can not adjust the current score higher since I lean to positive but still not that much.

---

> > > ### Author Response · Authors · 2022-08-08
> > > **Re: Response to authors**
> > >
> > > Hi Reviewer tbrQ, thanks for your response. We are glad that your concerns have been addressed and that you hope to see our paper at the conference. We note that a rating of 5 will not ensure that our paper will appear at the conference, so if you have no remaining concerns we respectfully ask that you consider raising your score. If there is anything else you would like us to change, add, or comment on, please let us know. Thanks again.

---

### Official Review · Reviewer_KMyy · 2022-07-12

**Rating:** 6
**Confidence:** 3
**Soundness:** 2 fair
**Presentation:** 3 good
**Contribution:** 2 fair

**Summary:**

**Main goal:** Come up with an evaluation methodology to evaluate compositional generalization (how consistently object attributes are expressed in representations obtained from test data) and use it to evaluate disentanglement and emergent language techniques to obtain insights about the representations that get learned.

**Summary of the proposed evaluation pipeline for measuring compositional generalization:**
  1. Use the training set to learn representations in an unsupervised manner (using e.g. variation methods, or emergent language teqhniques)
  2. Using a small portion of the training data annotated with object attributes, train a simple (e.g. linear) probe to extract the attributes from the representations.
  3. Check if the probe still works on unseen data.

**Findings:**
* If one commits to a linear probe to measure compositional generalization, then using bottleneck representations doesn't yield the best generalization results - using the layer before and after work better.
* Emergent language models yield better compositional generalization (when one uses a linear probe).
* Compositional generalization doesn't correlate well with other metrics that quantify disentanglement.


################

**Post rebuttal update** I thank the authors for their response. I've increased my score.

################

**Questions:**

* Learning the linear probe from limited training labels: I don't quite understand why one would use a limited number of training examples so train the probe. Isn't higher == better, as long as we don't mix test data?
* Line 52: Could you provide citations for the claim that today's SOTA unsupervised disentanglement models are built on top of variational generative models?
* Could you discuss the results you obtain with gradient boosted trees a bit more? Do they agree with what you get with a linear probe? What are the main disagreements?

**Limitations:**

A central limitation of the submission is that the proposed compositional generalization strategy isn't validation to fully capture compositional generalization. While high scores likely indicate strong compositional generalization, it's not clear whether low scores indicate lack of it. The paper would likely be stronger with sanity check experiments to validate the proposed eval methodology.

**Strengths And Weaknesses:**

STRENGTHS
**Interesting research direction:** Trying to directly measure compositional generalization is interesting - usually this skill is presumed in downstream evaluations. Coming up with a robust way of measuring this would be quite valuable.
**Useful signal for designing better techniques/algorithms:** As long as the proposed evaluation strategy is validated to genuinely capture compositional generalization, it can be a very useful tool in algorithm design by providing quantitative feedback.

WEAKNESSES
**Limitations of linear separability as a measure of compositionality:** Linear separability of object attributes (i.e. the crux of the proposed evaluation scheme) might not be the best way to capture how "compositional" a given representation is. It is perhaps not surprising that the bottleneck representations perform worse at this: representations in higher dimensions are easier to separate. I understand that the proposed metric measures more than just separability by evaluating the learned probe on a held-out test set. However, since linear separability is a necessary condition for this type of probe to generalize in the first place, I fear the conclusions drawn from this evaluation methodology might be a bit misleading.
**Affect of the probe model on results** This is a slightly general comment than the above one: how can we disentangle the inaccuracies caused by the probe itself form actual compositional generalization? This seems crucial to discuss here.
  * (Sanity check experiment) For example, if one generated a dataset of perfectly disentangled representations (not necessarily linearly separable) and used the proposed evaluation pipeline, would the proposed metric identify these representations as having 'high compositionality'? Having a sanity check experiment like this to test the proposed evaluation metric seems important.
  * (Another sanity check experiment) Another useful sanity check experiment could be checking how well the trained downstream probe works "in-distribution". That is, train the downstream probe with, say half the training data and evaluate it on the other half of the training data (not the test data). Does the probe generalize then? Are the results qualitatively different than if you evaluate on the unseen test set?
**Narrow evaluation:** It'd be interesting to see if the current takeaways hold for other types of models and datasets. For example, how about the Slot Attention model (https://arxiv.org/abs/2006.15055), on the CLEVR dataset?

---

> ### Author Response · Authors · 2022-08-02
> **Response to Reviewer KMyy Part2**
>
>
> Concern
>
> 3. *(A central limitation of the submission is that the proposed compositional generalization strategy isn't validation to fully capture compositional generalization. While high scores likely indicate strong compositional generalization, it's not clear whether low scores indicate lack of it.*
>
> What you identify is an important trade-off in the amount of supervision used when evaluating learned representations on downstream tasks. If we use stronger probes and more labeled data instead, one can also argue the opposite way: that the low score indicates a lack of it and it’s unclear if a high score means the generalization comes from the representations. For real-world applications, measuring how easy it is to get a better generalization from given representations is more meaningful. Given the sanity-checking experiments, we believe the linear and GBT probes are sufficiently capable of getting good generalization when the representations are good enough.
>
> Questions:
>
> 1. *Learning the linear probe from limited training labels: I don't quite understand why one would use a limited number of training examples so train the probe. Isn't higher == better, as long as we don't mix test data?*
>
>     It is true that using more labels may produce better performance. However, it would make the unsupervised representation learning step meaningless if a large number of labels are required to get good generalization on downstream tasks. In other words, with limited labels, we are able to measure how *easily** we get a downstream model with good generalization from the unsupervised learned representation. We discuss this point on Line 85-87 in the paper.
>
>
> 2. *Line 52: Could you provide citations for the claim that today's SOTA unsupervised disentanglement models are built on top of variational generative models?*
>
>     Yes, we will include citations for the SOTA models that rely on variational generative models.
>
> 3. *Could you discuss the results you obtain with gradient boosted trees a bit more? Do they agree with what you get with a linear probe? What are the main disagreements?*
>
>     The main observations in GBT results are consistent with linear probe results with below interesting observations:
>     - When using GBT probes, the performance of z_latent of VAE and EL models does improve but still underperforms z_pre and z_post.
>     - For the β = 0 disentanglement model, z_post performs well for both the regression and classification tasks, which is different from linear probes that favor z_pre in the regression task, .
>     - For the best performance for each learning model, GBT > linear probes in regression results while linear probes > GBT in classification results.

---

> ### Author Response · Authors · 2022-08-02
> **Response to Reviewer KMyy Part1**
>
> Thank you for the feedback and for recognizing our work as interesting and useful. We address your concerns and questions below.
>
> Concerns:
> 1. *Limitations of linear separability as a measure of compositionality.*
>
> First, we want to clarify that we are *not* trying to measure how “compositional” different representations are but rather to measure how easily we can achieve compositional generalization for downstream tasks with a given representation. We agree that linear evaluation has its own limitations, despite being widely used for evaluating unsupervised/self-supervised representation learning. Given these limitations, we included results using gradient-boosted tree (GBT) probes in the supplementary material. More discussions are given in the response to Q3 below.
>
> 2. *Affect of the probe model on results.*
>
> Thank you for suggesting the additional sanity check experiments. We agree that they would be an effective way of checking the capability of fitting the training data for selected probes. We ran the suggested experiments and summarize the results below.
>
> * In experiment 1, we test the oracle representations using the ground truth value of all attributes or the squared value of each attribute (which would not be perfectly linearly fittable). On the dSprites dataset, we use 500 samples to train linear or GBT probes for classification and regressions tasks. Their generalization performance is given in Table 1. It is expected that the attribute values can generalize perfectly with either a linear or GBT probe. However, linear probs can still fit the non-linear $attributes^2$ well. We think it may be due to the limited value range of attributes of our datasets. This experiment shows that if the learned representation is disentangled into attributes as people drive for, the linear head should not be a major issue to constrain the generalization performance.
>
> Table 1: Results of using the ground truth value of all attributes as representations.
> | Representations | Linear-Reg | Linear-CLS | GBT-Reg | GBT-CLS |
> |-----------------|------------|------------|---------|---------|
> | attributes      | 100%       | 100%       | 100%    | 100%    |
> | $attributes^2$  | 94.7%      | 100%       | 100%    | 100%    |
>
> * In experiment 2, the reviewer suggested testing the performance on the unlabeled part of the training set of the unsupervised learning stage (Unsup-Train). However, we don’t think it is strictly an “in-distribution” test but more a mixed scenario since the representation model is learned from them on the different  reconstruction task while unseen by downstream probes yet. Additionally, we test performance on the labeled samples for supervised training of probes (Sup-Train) that were seen by both the representation model and the probes. The results on the hold-out test set (Test) unseen by both training stages are given as a reference. In Table 2, we show results of evaluating the pre/latent/post representations of EL and beta-VAE(beta=0) with linear or GBT probes on the three sets in the dSprites dataset. The classification accuracy/ regression R2 score are given in each entry.
>    - We can see that for all representation models/modes and read-out probes, the performance of Unsup-Train and test is very close. It tells us that an example seen by the unusupervised pretraining stage does not necessarily have good performance in a downstream task in our setting.
>   - On the Sup-Train set, linear probes do not work well with latent representation of both VAE and EL models. Bad performance is expected for EL-latent since a linear probe is not a good choice for language-like messages as discussed in the paper. Combined with experiment #1, it indicates that VAE models do not produce representations that disentangle attributes.
>   - GBT probes fit EL-latent and VAE-latent well on Sup-Train and generalize poorly to Unsup-Train and Test. This further indicates that latent representations perform worse in providing good generalization than pre/post representations.
>
> Table 2: results on different data subsets.
> | Test set    | Probes  | 0-VAE-Pre   | 0-VAE-Latent | 0-VAE-Post  | EL-Pre      | EL-Latent   | EL-Post     |
> |-------------|---------|-------------|--------------|-------------|-------------|-------------|-------------|
> | Sup-Train   | Linear  | 99.93/94.48 | 73.27/64.83  | 100/95.99   | 100/92.55   | 46.13/13.35 | 100/99.39   |
> | Unsup-Train | Linear  | 84.83/84.15 | 70.89/63.13  | 98.74/71.29 | 91.02/84.7  | 39.11/9.92  | 99.99/98.04 |
> | Test        | Linear  | 84.3/83.9   | 70.98/63.1   | 97.88/73.98 | 90.56/84.6  | 38.8/9.5    | 99.94/97.85 |
> | Sup-Train   | GBT     | 100/96.44   | 99.33/93.64  | 100/98.81   | 100/95.86   | 96.93/83.63 | 100/99.83   |
> | Unsup-Train | GBT     | 77.67/79.44 | 76.8/77.57   | 97.73/88.34 | 81.58/81.02 | 56.19/58.53 | 99.53/95.83 |
> | Test        | GBT     | 76.76/78.95 | 76.09/77.26  | 96.83/87.82 | 80.86/90.56 | 54.84/57.36 | 99.52/95.68 |

---

### Official Review · Reviewer_JnAD · 2022-07-26

**Rating:** 8
**Confidence:** 4
**Soundness:** 4 excellent
**Presentation:** 3 good
**Contribution:** 4 excellent

**Summary:**

In this paper, the authors study the compositional generalization ability, i.e. unseen feature combination of elementary combinations, of beta-(TC)VAE and emergent language with Gumbel-softmax training. Both of the methods have been argued for their ability to disentangle features in the representation. However, the results on the two image datasets the authors did experiments on showed that the latent representations learned by beta-(TC)VAE do not generalize well and existing metrics show a different trend than their proposed metric, while emergent language is better and allowing flexible length and stochastic sampling are essential to the better performance.

**Questions:**

Have you done any study on the relationship between message lengths of instances and instance novelty? It would be very interesting to see the characteristics of language use. For example, for more novel combinations (less maximum overlap with training instances) maybe longer sequences should be used to describe.

**Limitations:**

Yes. The authors didn't discuss any negative societal impact, but I don't think there's any major negative societal impact in this paper that should be addressed.

**Strengths And Weaknesses:**

Strengths:

1. It is interesting to see that despite being argued for strong disentanglement, beta-VAEs still cannot generalize well compositionally with the heuristic regularizations. Pointing out the discrepancy between these two metrics is essential for researchers to understand the capability of these latent variable models.
2. It is also interesting to see that using flexible length discrete channel works better than the continuous channel.

Weaknesses:

I think the major weakness is the lack of discussion about evaluating compositional representation. Previous work, e.g. [1], has studied the latent structure of representations and methods to evaluate the compositional structure of them through simple compositional networks. I think this paper is in spirit similar to [1], and the difference and similarity should be addressed.

[1] Measuring compositionality in representation learning.

---

> ### Author Response · Authors · 2022-08-02
> **Response to Reviewer JnAD**
>
> Thank you very much for recognizing the importance of our work and its potential impact on the community.  Below is our response to your concerns and questions.
>
> 1. *The lack of discussion about evaluating compositional representation e.g. [1].*
> We discussed related work on measuring specific compositionality, e.g. disentanglement and emergent language in section 1. Thanks for pointing out [1] which is an important related work on compositionality. [1] proposed a compositionality metric (TRE) measuring how well a representation model can be approximated by an explicit compositional operator and learnable primitive representations.
> Similarity: we both explore the relationship between generalization compositionality metrics (TRE in [1] vs. disentanglement metrics and topographic similarity in our work).
> Difference: While [1] aims to come up with a more general compositionality metric, [1] admitted that some pre-commitment to a restricted composition function is essentially inevitable.  In contrast, we primarily focus on directly measuring compositional generalization, which connects much more closely to the motivations for work on compositional representation learning.
>
> 2. *Is any study done on the relationship between message lengths of instances and instance novelty?*
> Exploring the properties of the emergent language is not our primary focus since similar analysis exists in other EL works. In practice, we found that, after convergence, the EL model almost always uses the maximum message length on both the training and testing set. This is not surprising given that the reconstruction task drives the discrete message to be longer so that more information can pass through the discrete bottleneck.
>
> [1] Measuring compositionality in representation learning.

---

### Meta-Review · Area_Chair_CtN8 · 2022-08-30

**Recommendation:** Accept
**Confidence:** Certain

**Metareview:**

This paper investigates compositional generalisation (CG) of the representations learned from unsupervised learning through the lens of disentanglement and emergent language. They argue that models that are primed to disentangle do not learn representations that do well on CG and that models of emergent language do indeed perform well on CG. They further explore utility of EL by finding improved performance on plain generalisation and in learning from fewer labels.

The reviewers agree that the paper tackles an interesting and relevant problem, and the question and derived insights are valuable, and the experiments are quite thorough.
Where the reviewers raised valid concerns about evaluation, these were addressed by the authors in their rebuttal along with clarification on separability vs CG and additional experiments.

The only concerns that I believe could still be argued for here are that questions about the quality of representations could extend to models such as SimCLR/BYOL/MAE and that it would strengthen the paper a good deal if these were also discussed compared against.
To tighten the implied brief of the paper, it might be useful to rejig the title a bit---it currently comes across as exploring a spectrum between disentanglement and EL, whereas it's likely simpler to just state 'EL shows better CG than disentanglement as an objective' or something similar.

Overall, I believe this is a good paper, and should be accepted.

**Award:**

No

---

### Decision · Program_Chairs · 2022-09-14

Accept